Modeling ocean distributions and abundances of natural- and hatchery-origin Chinook salmon stocks with integrated genetic and tagging data

Jensen Alexander J. jensen.alex1502@gmail.com 1
Kelly Ryan P. 1
Satterthwaite William H. 2
Ward Eric J. 3
Moran Paul 3
Shelton Andrew Olaf 3
1 School of Marine and Environmental Affairs, University of Washington , Seattle , WA , United States of America
2 Fisheries Ecology Division, Southwest Fisheries Science Center, National Marine Fisheries Service, National Oceanic and Atmospheric Administration , Santa Cruz , CA , United States of America
3 Conservation Biology Division, Northwest Fisheries Science Center, National Marine Fisheries Service, National Oceanic and Atmospheric Administration , Seattle , WA , United States of America
Esteban María Ángeles
Electronic publication date: 2023 Nov 28
Publication date: 2023
Volume: 11
Electronic Location ID: e16487
Received 2023 Jul 27; Accepted 2023 Oct 27
Copyright year: 2023
Copyright holder: Jensen et al.
License: This is an open access article, free of all copyright, made available under the Creative Commons Public Domain Dedication. This work may be freely reproduced, distributed, transmitted, modified, built upon, or otherwise used by anyone for any lawful purpose.
License URL: https://creativecommons.org/publicdomain/zero/1.0/

Keywords: Chinook salmon, Genetic stock identification, Coded wire tag, Fisheries management, Marine life history, Distribution, Abundance, Integrated state-space model

Funding: University of Washington and National Oceanic and Atmospheric Administration (Northwest and Southwest Fisheries Science Centers) This work was supported by a post-doctoral fellowship, as a collaboration between the University of Washington and National Oceanic and Atmospheric Administration (Northwest and Southwest Fisheries Science Centers). The funders had no role in study design, data collection and analysis, decision to publish, or preparation of the manuscript.

==============================
Background

Considerable resources are spent to track fish movement in marine environments, often with the intent of estimating behavior, distribution, and abundance. Resulting data from these monitoring efforts, including tagging studies and genetic sampling, often can be siloed. For Pacific salmon in the Northeast Pacific Ocean, predominant data sources for fish monitoring are coded wire tags (CWTs) and genetic stock identification (GSI). Despite their complementary strengths and weaknesses in coverage and information content, the two data streams rarely have been integrated to inform Pacific salmon biology and management. Joint, or integrated, models can combine and contextualize multiple data sources in a single statistical framework to produce more robust estimates of fish populations.

Methods

We introduce and fit a comprehensive joint model that integrates data from CWT recoveries and GSI sampling to inform the marine life history of Chinook salmon stocks at spatial and temporal scales relevant to ongoing fisheries management efforts. In a departure from similar models based primarily on CWT recoveries, modeled stocks in the new framework encompass both hatchery- and natural-origin fish. We specifically model the spatial distribution and marine abundance of four distinct stocks with spawning locations in California and southern Oregon, one of which is listed under the U.S. Endangered Species Act.

Results

Using the joint model, we generated the most comprehensive estimates of marine distribution to date for all modeled Chinook salmon stocks, including historically data poor and low abundance stocks. Estimated marine distributions from the joint model were broadly similar to estimates from a simpler, CWT-only model but did suggest some differences in distribution in select seasons. Model output also included novel stock-, year-, and season-specific estimates of marine abundance. We observed and partially addressed several challenges in model convergence with the use of supplemental data sources and model constraints; similar difficulties are not unexpected with integrated modeling. We identify several options for improved data collection that could address issues in convergence and increase confidence in model estimates of abundance. We expect these model advances and results provide management-relevant biological insights, with the potential to inform future mixed-stock fisheries management efforts, as well as a foundation for more expansive and comprehensive analyses to follow.

Introduction

Various methods exist for tracking and estimating the distribution of fish in marine and freshwater environments (e.g., mark-recapture analyses based on tag recoveries, fisheries-independent and -dependent surveys, genetic stock identification), particularly for fish with multiple stock groupings of conservation importance. Analyses of resulting data, however, are often conducted in isolation and focus on data from a single monitoring method (Trudel et al., 2009; Bellinger et al., 2015; Shelton et al., 2019). Combining multiple sources of fishery information into joint (integrated) models can provide more comprehensive insights into all aspects of fish biology (e.g., life-history traits, fisheries reference points, distribution) (Maunder & Punt, 2013). Moreover, leveraging all available data to yield more robust demographic estimates is an efficient use of (often-public) resources that may lead to better management outcomes (Brownscombe et al., 2022), but such methods can be statistically complicated and require careful attention.

While integrated models have a long history in fisheries (reviewed in Maunder & Punt, 2013), they have become more common in the ecological literature and serve to link disparate sets of observations to estimate a common set of shared, often demographic parameters (Zipkin & Saunders, 2018). Examples from fisheries feature the integration of mark-recapture and snorkel survey count data (Staton et al., 2022), physical sampling and PIT tag antenna detection data (Conner et al., 2019), and fixed-location counts and acoustic telemetry data (Izzo, Zydlewski & Parrish, 2022). Other examples from the ecological literature include the integration of multiple data streams to estimate jackal abundance (Farr et al., 2021), population dynamics of greater horseshoe bats (Schaub et al., 2007), population trends of California spotted owls (Tempel, Peery & Gutierrez, 2014), viability of Great Lakes piping plovers (Saunders, Cuthbert & Zipkin, 2018), and vital rates of North American waterfowl (Arnold et al., 2018). The benefits of combining datasets for common inference are significant—for example, integrated models can reduce uncertainty (where constituent datasets agree), and better represent the nuances that the strengths of different data sources reveal. However, these benefits come at the cost of increased model complexity, demands on the expertise of the analyst, and structural demands of the models themselves (Maunder & Punt, 2013). Consequently, where datasets reflect fundamentally different aspects of a system and model processes do not sufficiently capture these differences (i.e., the model is misspecified), joint models may result in biased model estimates and fail to capture meaningful information about the world (Maunder & Piner, 2017).

Because of their cultural, ecological, and economic value in the Northeast Pacific Ocean, Pacific salmon (Oncorhynchus spp.) species have been subject to a long history of intensive research and monitoring. However, the complexities of these species’ life histories over an enormous spatial range (i.e., 1000s of kms, spanning multiple geopolitical boundaries) and the often-disparate methods of surveying mean that different datasets frequently reflect a localized understanding of phenomena that play out at larger scales. Historically, Chinook salmon (O. tshawytscha) and coho salmon (O. kisutch) have been targeted by recreational, commercial, and tribal fisheries catching a mix of stocks, each arising from different river systems. In addition to their fishery importance, increased research focus has been placed on specific stocks over the last two decades, as multiple populations of Chinook and coho salmon have been listed under the U.S. Endangered Species Act (ESA) and Canadian Species at Risk Act (SARA). Chinook and coho salmon have been monitored, managed, and analyzed using coded wire tags (CWTs), in which juvenile fish have wire-based tags with specific alphanumeric codes implanted in their nasal cavity prior to or during their migration from freshwater to the ocean. CWTs have been deployed by state, federal, and tribal management agencies in the USA and Canada using a standardized protocol across the Northeast Pacific Ocean for decades (i.e., since the 1970s). They offer precise assignment of recovered fish to natal region and age, and while their use has been largely restricted to hatchery-produced fish (Bernard & Clark, 1996), many stocks do have CWT programs. Data from CWTs support large scale fishery-management models in the Northeast Pacific Ocean (e.g., Pacific Fishery Management Council, 2008; Pacific Salmon Commission, 2023), and large-scale analyses have successfully employed historical patterns of CWT recoveries to reconstruct marine distributions (Weitkamp, 2010; Shelton et al., 2019; Shelton et al., 2021) and productivity (e.g., Sharma et al., 2013; Kilduff, Botsford & Teo, 2014; Welch, Porter & Rechisky, 2021) of Chinook salmon stocks.

Genetic stock identification (GSI) has increasingly become available for Pacific salmon, including Chinook salmon, and sampling and laboratory efforts starting in the 1980s have yielded numerous datasets for the species. Genetic stock identification is a general term encompassing a variety of methods of genetic analysis, including the use of markers such as allozymes, mitochondrial DNA, microsatellites, and single nucleotide polymorphisms, with the common aim of distinguishing stocks or population units (Grant et al., 1980; Utter & Ryman, 1993; Shaklee et al., 1999; Cronin et al., 1993; Banks et al., 2000; Narum et al., 2008). Individuals from distinct source populations of anadromous fish species, including Pacific salmon, are intermixed in the ocean, and GSI is a means to identify the natal origin of those fish and inform management of populations with varying levels of conservation concern. Although GSI frequently has been applied to salmonid fish species (Flannery et al., 2010; Dann et al., 2013; Satterthwaite et al., 2015; Jensen et al., 2021; Beacham et al., 2022), the same techniques are applied to varied non-salmonid and non-fish species (Bolker et al., 2007; Hasselman et al., 2016; Scribner et al., 2022; Henriksson et al., 2023). A strength of GSI data for species with strong genetic differentiation among populations, relative to tagging-based identification alternatives, is that it can be used to identify the origin of individuals sampled without a reliance on tagged, and typically hatchery, fish and therefore potentially improve information about populations that have not been tagged (e.g., wild salmon populations of conservation concern) (Hess et al., 2014; Satterthwaite et al., 2014).

As both CWT and GSI have been used separately to inform the relative abundance and distribution of salmon stocks for decades, it is surprising that there have been no completed, large-scale efforts to formally combine information from CWT and GSI and inform both of these stock attributes (see Pacific Salmon Commission (2007) for proposed analyses). Reported integrations of CWT and GSI data largely have been confined to regional applications informing escapement or fishery-specific harvest (Korman et al., 2011; Bernard et al., 2014; Barclay et al., 2019). Here we present a first attempt to leverage the relative strengths of each data type by building joint models that can utilize both data types and endeavor to improve estimates of marine abundance and distribution for a small number of Chinook salmon stocks. During model construction and testing, we encountered a number of significant technical challenges that we were unable to entirely and satisfactorily overcome. Such challenges are not unexpected in constructing complicated integrated models, and we describe both the successes and failures encountered during model building and testing. Importantly, we describe additional data needs and assumptions that will be necessary to smoothly integrate CWT and GSI information in the future. We view this paper as a roadmap that can be used to help guide future investigators interested in advancing the use of CWT and GSI for salmon biology and management.

To understand how linking GSI and CWT data can improve estimates of ocean distribution and abundance, we construct our joint CWT and GSI model using Chinook salmon data for four, fine-resolution stock groupings from California and southern Oregon with particular management and conservation relevance. These four stocks represent a range of CWT information availability (from sparse to abundant) and provide a venue for understanding the relative value of adding GSI information to existing CWT-based models. CWT and GSI datasets for these stocks previously have only been analyzed separately (Satterthwaite et al., 2013; Bellinger et al., 2015; Satterthwaite et al., 2015). We build upon previous state-space modeling efforts that relied primarily on CWT recoveries (Shelton et al., 2019; Shelton et al., 2021) and add GSI information to estimate the marine spatial distributions for both hatchery- and natural-origin fish from the four selected Chinook salmon stocks. Model results provide refined estimates of spatial distribution for selected stock groupings, novel estimates of stock abundances over time, and insights into the strengths and weaknesses of conducting this type of joint modeling. We expect this joint model provides a foundation for more expansive analyses that include more stocks, life history types, and datasets. Understanding the ocean spatial distribution and abundance of multiple stocks simultaneously is valuable to informing mixed-stock fisheries management to target healthy stocks while avoiding weak ones (O’Farrell & Satterthwaite, 2015), understanding spatial portfolio effects and dynamics in abundance for different ocean areas (Sullaway, Shelton & Samhouri, 2021), and quantifying trophic interactions, including the prey needs of charismatic megafauna (Stewart et al., 2021).

Materials & Methods

Methods overview

We aim to provide the first model to formally integrate information from (1) large-scale, long-term CWT programs and (2) more recently developed monitoring based on GSI sampling of fisheries catches, to derive estimates of marine distribution and abundance for Chinook salmon. We first describe the CWT-based model structure developed in Shelton et al. (2019) and Shelton et al. (2021) before describing the additional GSI-associated components, including incorporation of GSI mixture observations, stock-specific age structure, freshwater run size, and total marine fisheries catches. Several of these components act as model constraints to address challenges in integrating CWT and GSI data sources into a single, joint model.

CWT-based (CWT-only) model structure

We first present the model structure and data of the CWT-based state-space model, described in detail in Shelton et al. (2021), which served as the foundation for subsequently integrating GSI data. In this model, we estimate the spatio-temporal dynamics of tagged releases of Chinook salmon from distinct stocks, using a state-space model that distinguishes biological processes from population observations. We track each CWT release group (i.e., juvenile salmon reared in a hatchery and released together) from initial release abundance (ranging between 9,000 and 3.5 million) through final escapement for spawning. We describe the corresponding age classification schedule in Table S1.1. Patterns in CWT recoveries from fisheries-dependent and fisheries-independent sampling in marine and freshwater environments provide information on four biological processes: (a) the mortality of fish prior to spring of age 2; (b) fishing mortality by age and ocean region among fleets; (c) spatial distributions of stocks among discrete ocean regions; and (d) age-specific departure of fish from the ocean due to maturation and spawning. The model compares model-predicted CWT recoveries to observed recoveries across releases, times, and regions to find parameters that minimize the distance between predictions and observations.

In a departure from previous modeling efforts (Shelton et al., 2019; Shelton et al., 2021), we reduced the number of modeled stocks, releases, and ocean regions to facilitate the subsequent integration of GSI data and focus comparisons across stocks of primary interest. We followed Shelton et al. (2019) in assuming spatial distributions of stocks did not vary annually. We tracked the population dynamics of 284 tagged groups of Chinook salmon from between 1979 and 2018, which resulted in 82,542 observations of CWT catch (Table S2.1). The selected four stocks were fall-run Chinook salmon from California’s Central Valley (code: SFB), coastal rivers and streams south of Klamath River to the Russian River (CAC), Klamath and Trinity Rivers (KLT), and coastal rivers and streams in northern California and southern Oregon, south of the Elk River and north of Klamath River (NCASOR). These stocks reflect finer resolution groupings than those specified in previous analyses and largely map onto the structure of both managed Evolutionarily Significant Units(ESUs) and baseline reporting groups used in GSI (Seeb et al., 2007; Clemento et al., 2014; Shelton et al., 2019; Shelton et al., 2021). We modeled distribution along the U.S. west coast, divided into 8 ocean regions (i.e., Monterey (MONT), San Francisco Bay (SFB), Mendocino (MEN), Northern California (NCA), Southern Oregon (SOR), Northern Oregon (NOR), Columbia River (COL), and Washington (WAC); see Fig. 1) due to limited availability of GSI data north of WAC; we rarely observed CWT recoveries for the four focal stocks north of WAC (Table S2.5; see also Shelton et al., 2019; Shelton et al., 2021). We incorporated three additional years of CWT recovery data compared to previous analyses (2016–2018; Shelton et al., 2021) to maximize overlap between the temporal coverage of CWT and GSI sampling (Fig. 2). Finally, we changed the definition of seasons spring and winter to better match patterns of fishery harvest (Supplemental Information S1). We implemented the statistical model in Stan (Stan Development Team, 2022) as structured in R (rstan v2.21.2; R Core Team, 2023; Stan Development Team, 2023). Full details on model structure and data for this and other portions of the model are provided in Supplemental Information S1.

Figure 1 Map of the study area, discrete coastal regions, and some natal spawning rivers of modeled Chinook salmon stocks.

The map is unique, was not reproduced from existing, copyrighted geospatial data or publications, and the background hillshaded bathymetry in the main panel was sourced from NOAA National Centers for Environmental Information (2022).

Figure 2 Summary of CWT and GSI data observations, based on presence/absence of data only, by stock, region, season, and year.

Integrated CWT- and GSI-based (CWT+GSI) model structure

In addition to tracking the abundance and distribution of focal stocks using hatchery-derived CWT release groups, we separately track the abundance and distribution of total populations from each of the focal stocks (i.e., including hatchery- and natural-origin fish). More specifically, we treated and tracked all fish “released” (i.e., out-migrated) into the marine environment in a given year and from a given stock as a single total release group. Tracking total release groups is necessary for us to link model processes to GSI data because GSI provides information on the composition of all fish, not just tagged fish. These new estimates also expand our inference to provide insights into the abundance of a given stock at a given time and place. We only track these total release groups starting at the spring of age 2 (i.e., after any juvenile mortality occurs following initial entry into the ocean), as we lack sufficient data on total out-migrating smolt abundances. Estimation of life history processes for the total release groups is based in part on GSI data to inform relative stock composition in marine fisheries. Integrating GSI data into the model simultaneously estimates shared life history parameters for CWT and total release groups and provides estimates of spatial distribution and abundance reflective of fish both with and without CWTs.

We incorporated individual assignment data from dockside GSI sampling of recreational landings from California in 1998–2002 and GSI sampling of commercial catch from California, Oregon, and Washington in 2006-2014 into the model to inform the abundance and distribution of total release groups; GSI sampling from commercial fisheries relied on voluntary participation and data collection by fishermen (Satterthwaite et al., 2014; Bellinger et al., 2015; Satterthwaite et al., 2015; Fig. 2; Supplemental Information S2). Our use of individual assignment probabilities, rather than raw genotype data, to generate GSI observations was intended to reduce computation requirements and facilitate use of publicly available GSI datasets; however, we emphasize that we used all individual probability-of-group-membership data for each sampled fish, and not just maximum assignment probability, to generate unbiased estimates of stock composition (Koljonen, Pella & Masuda, 2005). In total, we included GSI data spanning 139 unique region-season combinations for two fisheries (i.e., commercial troll, recreational), including a total of 63,018 Chinook salmon sampled for GSI. We selected these datasets based on their spatial and temporal coverage and available sample sizes for characterizing fishery composition. Relative availability of GSI and CWT data varies among stocks; for stocks with more extensive CWT tagging efforts like SFB, CWT observations greatly outnumber GSI observations, while there were similar numbers of CWT and GSI observations for rarer and less supplemented stocks like CAC (Fig. 2).

Similar to our treatment of CWT data, we compared predicted patterns of stock-specific fishery catch to observed stock compositions from GSI sampling. However, in contrast to CWT observations that provide information on the specific age of recovered fish, GSI data only provide information on relative contributions of stocks to the mixture of sampled fish (e.g., approximately 12 of 200 sampled fish in a given region, season, and year are assigned to stock SFB). Furthermore, GSI data can include genetic assignments for stocks other than the four focal stocks (i.e., non-focal stocks like Columbia River Chinook salmon stocks); these can contribute substantially to GSI sampling in ocean regions progressively further north. Our model structure accounts for these complexities. We also note that the GSI reporting group for SFB includes Feather River spring-run fish along with late-fall fish and the KLT reporting group includes both Klamath River fall- and spring-run fish, due to limitations in the available GSI baselines (Seeb et al., 2007; Clemento et al., 2014). Therefore, to an extent, GSI information will reflect information on more than strictly fall-run fish for these two stocks. However, fall-run is numerically dominant for both stocks (Pacific Fishery Management Council, 2020).

To incorporate GSI mixture observations in our model structure, we compare the expected proportional contributions of each focal stock to fisheries harvest to observed GSI data using a zero-and-one inflated Dirichlet regression model with estimated overdispersion (Jensen et al., 2022). This model structure allows for zero and non-integer GSI observations of focal stocks—that is, instances in which none or all the observed fish derive from a given stock—in addition to an overdispersion term that implicitly allows the model to weight the importance of GSI data relative to competing data sources, including CWT data (i.e., greater overdispersion, as measured by a smaller overdispersion parameter, indicates reduced model weighting of GSI data).

To calculate expected proportional contributions of each focal stock to harvest, we first obtain expected catches for a given stock r, region l, season s, year c, and gear g (μ1,l,s,c,r,g) by summing expected catches across total release groups representing all available ages. Release group-specific expected catches already are estimated in the model as a function of group abundance, spatial distribution, and fishery mortality rate. We use the total number of GSI sampled fish and summed number of fish assigned to focal stocks F using GSI to estimate the expected proportion of sampled catch comprised of focal stocks (p2,l,s,c,F,g), then estimate the total expected catch across all stocks μ1,l,s,c,g by dividing the expected catch summed across all focal stocks (μ1,l,s,c,F,g) by this expected proportion (μ1,l,s,c,g = μ1,l,s,c,F,g/p2,l,s,c,F,g). This expansion is necessary to account for the fact that we only model catches for four focal stocks but total fisheries catch often includes additional stocks. Finally, we estimate the expected proportional contribution of each focal stock r to total harvest by dividing expected stock-specific catch μ1,l,s,c,r,g by μ1,l,s,c,g.

Because the complete model struggled to converge, we implemented several additional model constraints. In the absence of release abundance data for total release groups (e.g., number of out-migrating smolts in a given year), we constrained estimates of total release abundances to biologically feasible values using the following data: (1) annual estimates of stock-specific run size (i.e., the number of mature adults that enter freshwater to spawn) when available, (2) estimates of total fishery landings by season, region, and fishery type, and (3) within-model estimates of CWT release group abundances. First, using total release groups within the model, we obtained the expected number of fish entering freshwater across all ages for each stock in a given stock and year; we then minimized the distance between these numbers and independent estimates of run size generated from assorted survey data using a lognormal likelihood with fixed standard deviation values (Supplemental Information S2). Second, we compared total expected catch for recreational and commercial fisheries for all seasons and regions from 1998–2014 (i.e., μ1,l,s,c,g) to corresponding independent estimates of landings using a normal likelihood and a fixed coefficient of variation (Pacific Fishery Management Council, 2019). To deal with seasons, regions, and years without the corresponding GSI data necessary to estimate total expected catch (i.e., no data to estimate the proportion of focal stocks in expected catch, p2,l,s,c,F,g), we estimated these proportions hierarchically using a logit-normal likelihood, expected mean proportions of focal stocks in each region, and an estimated standard deviation. The hierarchical structure facilitates borrowing of information from seasons, regions, and years with the necessary GSI data to directly estimate p2,l,s,c,F,g. Third, to further constrain estimates of abundance and improve model convergence, we specify the abundance of total release groups as the sum of the expected number of CWT-tagged fish (i.e., already estimated internally in the model and informed by known CWT release sizes and CWT fisheries recoveries) and the modeled number of non-CWT-tagged fish (i.e., including both untagged hatchery- and natural-origin fish); this functionally establishes the CWT abundance as the lower bound of total abundance and reduces the occurrence of unrealistically low abundance estimates for some release groups. Finally, the model allows process variability in estimates of CWT and total release group abundances: we constrained the process variability to be identical for every release (i.e., either CWT or total) associated with the same stock and brood year, based on the implicit assumption that effects of population drivers not included in our model are shared by releases from the same stock and brood year.

Exploration of synthetic age structure (CWT+GSI+Age)

Based on persistent challenges in achieving model convergence for estimates of total release group abundances for the CWT+GSI model (see Results), we added an exploratory set of age-structure constraints to the model by providing synthetic samples of ages. Specifically, for a given season, year, and stock, we provide counts of aged fish, generated from the product of fixed age proportions (i.e., either ages 2–5 or 3–5, depending on season) and a specified number of sampled fish, as part of a new multinomial likelihood below. A.,s,c,r∼Multinomialp3,.,s,c,r.

Specifically, counts of aged fish for each season s, year c, and stock r, or A.,s,c,r, are used as a form of prior to constrain the proportional contributions of each release age to the total stock abundance, or p3,.,s,c,r. We provide artificial counts of aged fish due to a lack of systematic data collection of age structure for catches in the marine environment. We estimated fixed age proportions using reconstructed total Klamath River Fall Chinook ocean abundances (Pacific Fishery Management Council, 2021) that we assumed applied to CAC and NCASOR as well; we provide separate proportions for SFB based on both expectations of skew to younger ages and expert judgment (Carvalho et al., 2023). We recognize these proportions are coarse, imprecise values, and are primarily intended to demonstrate the value of improved data availability in the future. Furthermore, although the inclusion of CWT data in the KLT cohort reconstruction, alongside our use of CWT recoveries in this model framework, could represent double-weighting of these data to some extent, we note that a large fraction of age data for the reconstruction was obtained from unmarked fish. We anticipated this synthetic age structure would constrain model flexibility in estimating abundances of total release groups and subsequently improve model convergence.

Model analyses

We first ran the model with only CWT-based catch observations (CWT-only) using the previously described number of stocks, release groups, regions, and years of data, and we characterized estimates of seasonal ocean distribution for each stock. We then ran the model with both CWT- and GSI-based catch observations, but no synthetic age structure (CWT+GSI), and characterized both estimates of seasonal ocean distributions and total abundances for each stock. Finally, we ran the model with CWT- and GSI-based catch observations, in addition to synthetic age structure (CWT+GSI+Age), and characterized improvements in model convergence in addition to any changes in estimates of seasonal ocean distribution and abundance.

We ran each model with six chains, 400 burn-in iterations, and 1,400 retained iterations. We primarily characterized model convergence using R-hat (Gelman & Rubin, 1992; Vehtari et al., 2020).

Statement on data availability

Raw and processed data, including CWT and GSI catch observations, estimates of run size, and estimates of landings, in addition to the corresponding processing scripts, are available at Zenodo (DOI 10.5281/zenodo.8057794) or by contacting the authors.

Results

GSI observations

In the CWT+GSI and CWT+GSI+Age models, we included GSI observations from every region, years 1998–2002, 2006–2007, and 2009–2014, and spring, summer, and fall seasons across both commercial troll and recreational hook-and-line fisheries (Supplemental Information S2). Lack of consistent fishing effort in winter resulted in a lack of winter-based GSI data, and observations from the more northern regions COL and WAC were relatively rare. For the GSI observations from the commercial troll fishery (Fig. 3), the SFB stock generally predominated observed catch, particularly in southern spatial regions. In some seasons and years, the KLT stock contributed the most to sampled catch among focal stocks; this generally occurred in either the NCA or SOR regions. The total contribution of the four focal stocks to sampled catch generally decreased with increasing latitude, consistent with their spawning locations. We present GSI observations from the recreational fishery in Fig. S2.12; these observations were restricted to California regions, years 1998–2002, and were again predominated by the SFB stock.

Figure 3 Summary of GSI data from commercial troll fisheries.

Points represented observed mixture proportions by stock for each combination of spatial region, season, and year. Point size (N) indicates the total number of sampled fish within each stratum.

CWT-based (CWT-only) model

We observed satisfactory model convergence for the CWT-only model (R ˆ<1.01 for all parameters). Detailed model diagnostics for this and subsequent models are shared in Supplemental Information S3.

We present estimates of seasonal spatial distribution for the four focal stocks, including 95% credible intervals, in Fig. 4. The estimated spatial distributions of SFB correspond well to those estimated in Shelton et al. (2019) and Shelton et al. (2021), as expected, but also to other qualitative estimates of ocean distribution (e.g., Weitkamp, 2010; Satterthwaite et al., 2013; Bellinger et al., 2015; Satterthwaite et al., 2015). This stock generally occurs between approximately Morro Bay (CA) and Cape Falcon (OR), and is particularly concentrated in the MONT, SFB, and MEN regions. The spatial distribution estimated for CAC is the most comprehensive one for this stock to date, based on spatial extent and the amount of utilized data, and exhibits a more northerly distribution than SFB, predominantly occurring in the MEN and NCA regions. Estimated distributions for KLT are concentrated more northerly still, with the greatest proportion of the stock generally occurring in the NCA region. Finally, the NCASOR stock had the most northerly distribution, with the greatest proportion of the stock occurring in the NCA and SOR regions; the estimated distribution also is the most comprehensive to date for this stock. Estimates of spatial distribution for CAC and NCASOR have wider credible intervals than those estimated for SFB and KLT, consistent with the lower data availability for these stocks. There were minor but inconsistent differences in seasonal distributions for each stock, further supporting the inclusion of seasonality in the model structure.

Figure 4 Estimated spatial distributions of the four modeled stocks by season.

Shades of gray in each facet represent the total number of recovered CWTs that contributed to estimates of spatial distribution. Error bars represent 95% credible intervals.

CWT- and GSI-based (CWT+GSI) model

The CWT+GSI model exhibited some challenges in model convergence (median R ˆ=1.01, R ˆ<1.01 and R ˆ<1.1 for 48% and 88% of tracked parameters). Most issues in model convergence were associated with new estimates of abundance for total release groups (Table S3.1). In particular, the initial abundance at model age 1, before application of fishing or natural mortality, were not fully estimable; there are many combinations of abundances for individual cohorts that can produce abundances similar to the GSI observations (see Supplemental Information S3 and below for more details and discussion).

Estimated ocean distributions for the CWT-only and CWT+GSI models were generally similar, with increasingly northerly distributions for stocks with more northerly spawning distributions (Fig. 4). However, ocean distributions differed between models for some stocks and seasons. For SFB, the CWT+GSI model estimated a more southerly distribution in winter-spring and lower occurrence in MONT in fall, based on non-overlapping 95% credible intervals, relative to the CWT-only model; all other estimated distributions had overlapping intervals. Similarly, although all credible intervals overlapped for CAC, the CWT+GSI model indicated a more southerly distribution in winter-spring and summer and a more northerly distribution in fall. Estimated spatial distributions were highly similar among models for KLT, as all credible intervals again overlapped. For NCASOR, the CWT+GSI model indicated ocean distributions were generally more diffuse but did not show any consistent shifts to either the north or south. Credible intervals for NCASOR did not overlap between models for several regions in both summer and fall. The width of credible intervals did not differ appreciably between the CWT-only and CWT+GSI models.

New parameter estimates from the CWT+GSI model include total estimates of stock abundance over time. Because we estimated abundances of total release groups over time, we also were able to calculate estimates of total stock abundance over time for years 1998–2014 (i.e., abundances of all tracked ages). We present example estimates of regional and seasonal stock abundances for 2002 and 2009 (Fig. 5). These years represented convenient example years with very different estimates of stock contributions and total abundance to illustrate the value of these new parameter estimates. Estimated stock abundances in 2002 were high, exceeding 800,000 fish across focal stocks for some regions in spring and summer, and were dominated by the SFB stock. In 2009, estimated stock abundances were noticeably lower and more heterogeneous across stocks. Relative distributions of total fish abundance among regions also varied by year as a function of proportional stock abundances; fish were distributed more northerly overall when northern origin stocks like KLT were predominant. Although not shown, we also can present estimates of uncertainty in stock abundances via 95% credible intervals as part of standard model output. These estimates represent novel historical estimates of relative and total abundances for an assemblage of both rare and abundant Chinook salmon stocks, with the obvious caveat that the model struggled to converge in estimating these values.

Figure 5 Estimated stock abundances and proportional contributions to total abundance across focal stocks by season, year, and region for the CWT+GSI model.

We selected 2002 and 2009 to highlight contrast in total abundance and stock-specific proportional contributions to abundance.

The CWT+GSI model also generated some model parameter estimates that differed noticeably from those estimated by the CWT-only model. Specifically, the CWT+GSI model estimated higher juvenile mortality rates and fishery mortality rates than CWT-only, with subsequent effects on estimates of CWT and total release group abundances over time (Supplemental Information S3). This shift in estimated mortality rates shows instability in estimation of these mortality terms as we add data and model complexity, and suggests that further investigation may be required to identify which set of mortality rates is most defensible.

Synthetic age structure (CWT+GSI+Age) model

We added synthetic age structure to the CWT+GSI model in an attempt to improve model convergence, particularly for estimates of total release group abundances over time. The CWT+GSI+Age model still exhibited challenges in overall model convergence (median R ˆ=1.01, R ˆ<1.01 and R ˆ<1.1 for 49% and 90% of tracked parameters). Addition of age structure-related model constraints did improve convergence for estimates of abundance for total release groups, as well as other parameters, but estimates of initial abundance at model age 1 before application of fishing or natural mortality again failed to converge (Table S3.2). The fact that the synthetic age structure did not resolve all issues in model convergence is unsurprising, based on our specification of fixed age proportions from a KLT-based cohort reconstruction. Failure to improve convergence for initial total release group abundances indicates additional data or model constraints still are required to resolve these parameter estimates.

Estimates of ocean distribution for focal stocks did not differ appreciably between the CWT+GSI and CWT+GSI+Age models, suggesting that difficulty in model convergence for some GSI-based parameters did not substantially affect estimates of spatial distribution for models that include GSI-based components (Fig. S3.26). Additionally, estimates of total stock abundances by region, season, and year also did not differ substantially (Fig. S3.27). The only parameter we observed to change with the inclusion of synthetic data was the overdispersion term for GSI, which indicated less weight is applied to GSI data for the CWT+GSI+Age model.

Discussion

The primary objective of our analysis was to develop a novel integrated statistical model to reconcile several fisheries dependent data sources (CWT, GSI) that have been used to inform the spatial distribution of Chinook salmon populations in the Northeast Pacific Ocean. Our model using only CWT data converged and provides the most comprehensive estimates of marine distribution for Central Valley, California Coastal, Klamath, and North California and Southern Oregon fall-run Chinook stocks to date, based on our inclusion of CWT data from new release years and hatcheries. Estimated spatial distributions generally matched expectations based on spawning distributions and earlier modeling efforts (e.g., Weitkamp, 2010; Shelton et al., 2019), and uncertainty in estimated distributions reflected available sample sizes of recovered CWTs. Our integrated model that combined GSI data with CWT data appeared to be more problematic, however. In a variety of sensitivity analyses designed to constrain parameters or model assumptions (e.g., using estimates of run size, commercial and recreational landings), our joint model was not able to converge. Parameters that remained particularly challenging to estimate were total release group abundances. Because total release group abundances are only informed by GSI data (i.e., CWT data only inform the abundance of a known number of tagged fish and tagged fish make up an unknown fraction of the total number fish) and GSI does not provide observations of age structure, there are many possible combinations of total release abundances that can yield similar aggregate catches to those observed via GSI in commercial and recreational fisheries. When we added age-structure constraints to our model with synthetic data (CWT+GSI+Age model), we observed generally improved model fits but the model still did not result in satisfactory convergence. The inability of the synthetic age structure to entirely resolve convergence issues is not entirely unsurprising because the age information may conflict with true time-varying age distributions; however, improvement in fitting with the inclusion of age structure suggests that pairing age and GSI information may be necessary to allow for estimation of salmon abundance for individual Chinook salmon cohorts. Both the CWT+GSI and CWT+GSI+Age models produced similar estimates of marine spatial distribution and abundance, suggesting that the inclusion of age structure information does not substantially change information about distribution. Integrated estimates of spatial distribution and abundance with uncertainty, including those presented here, are relatively novel for Chinook salmon and can inform more effective mixed-stock fisheries management and conservation efforts.

We observed several differences in estimated spatial distribution between the models excluding and including GSI data. Models with GSI data showed a general trend of estimating greater proportional occurrence of stocks in southern regions (i.e., MEN, SFB, MONT) in winter-spring and summer; trends were less clear in fall. One possible explanation for the shift in estimated marine distribution is that untagged natural- and tagged hatchery-origin fish exhibit different persistent marine distributions and accounting for natural-origin fish with GSI data partially captures these differences. This explanation may contrast prior observations of limited differences in coarse-scale stock composition estimated using either GSI or CWTs (Weitkamp, 2010; Sharma & Quinn, 2012; Satterthwaite et al., 2015; Satterthwaite & O’Farrell, 2018), although differences in distributions among proximate populations at finer resolutions suggest single population indicators of distribution may not completely represent stock dynamics (Beacham et al., 2020; Freshwater et al., 2021). Alternatively, the inconsistent temporal coverage of GSI data, relative to the broader coverage for CWT data, means that GSI sampling could have occurred during periods of time in which ocean distributions were generally more southerly. The magnitude of the differences in spatial distribution is similar to temperature-based differences in estimated spatial distribution for SFB between 1997 and 2008 (Shelton et al., 2021). Accounting for interannual variability by including environmental conditions in modeling, similar to the inclusion of temperature effects in Shelton et al. (2021), could help resolve the probable cause for the shifted distributions. The fisheries responsible for generating the current GSI data also could have influenced distribution estimates. Commercial and recreational fisheries can produce different estimates of stock composition and distribution, as commercial fisheries frequently operate in deeper water and at greater distances from ports (Satterthwaite & O’Farrell, 2018). We expect locating and adding new sources of GSI data from different time periods, fisheries, and spatial locations can inform how much variability in estimates of spatial distribution may be attributed to coverage of GSI data. Finally, we note directional biases in GSI assignments (e.g., mis-assigned fish from one group preferentially are assigned to another group), in addition to the necessary inclusion of low abundance spring-run groups in some of the presumably fall-run stocks, have the potential to influence model results. GSI-based estimates of KLT distribution in particular may not completely capture the distribution of fall-run Klamath River fish specifically, given estimated differences in fine-scale distribution between spring- and fall-run fish (Satterthwaite & O’Farrell, 2018); however, our estimates of KLT distribution were more consistent between model versions than those for other stocks. For SFB, the inclusion of more southerly-distributed late-fall Chinook (Satterthwaite et al., 2013) might affect the distribution estimated compared to fall run alone, though fall run greatly outnumbers late fall (Pacific Fishery Management Council, 2020).

New estimates of life history characteristics, including fishery mortality rates, release-specific juvenile mortality rates, and population-specific spatial distribution, ocean abundance, and spawner run size, represent important contributions to ongoing Chinook salmon fisheries management. The most comprehensive estimates of spatial distribution to date for the four modeled stocks represent varying improvement from previous understanding, with the greatest improvement for rarer stocks with limited marking and tagging programs (i.e., CAC and NCASOR). Previous estimates of distribution for all stocks include relative patterns in CWT recoveries prior to 2005 (Weitkamp, 2010), and a subset of CWT recoveries from our focal stocks (i.e., SFB, CAC, KLT, NCASOR) were included in previous distribution modeling (Shelton et al., 2019; Shelton et al., 2021). Previous estimates of CAC and KLT distribution in California and Oregon waters were generated using GSI in years 2010 and 2011 (Satterthwaite et al., 2014), estimates of SFB distribution were generated using CWT recoveries between 1983 and 2007 (Satterthwaite et al., 2013), and distributions of fall-run and spring-run Klamath River fish were generated using CWT recoveries between 1983 and 1989 (Satterthwaite & O’Farrell, 2018). Additionally, the integrated model allows estimation of stocks’ ocean abundance by season, year, and region between 1998 and 2014, informed by numerous sources of empirical data including fisheries landings, CWT recoveries, GSI sampling, and run size estimates. Estimated abundances exhibited extensive variability in stock abundances and composition over time, in accordance with similar studies that utilized either GSI or CWT data to generate estimates of stock composition and/or standardized catch rates and inform abundances and distributions of Chinook salmon (e.g., Satterthwaite et al., 2015; Moran et al., 2018; Freshwater et al., 2021). These estimates of abundance, paired with estimates of distribution, offer the potential for improvement in mixed-stock fisheries management in Pacific Northwest, particularly for the spatial allocation of fishing effort to target healthy stocks while avoiding more vulnerable stocks (e.g., O’Farrell & Satterthwaite, 2015). Finally, we again note that adding GSI data to these models shifted the estimates of juvenile and fishery mortality rates, and this uncertainty has implications for resolving potential bottlenecks in population productivity. For example, increased juvenile and fishery mortality rates in the models with GSI data necessarily correspond with higher starting ocean abundances of stock release groups and suggest ocean mortality could play a larger role in regulating population trends relative to early life history processes in freshwater. However, we emphasize we did not have the necessary data to resolve which set of mortality estimates is best supported.

These novel, direct estimates of ocean abundances have the potential to inform fisheries management. Current estimates of stock abundance for use in fisheries management and planning for many stocks are generated by the Pacific Fishery Management Council (PFMC) using the Fishery Regulation Assessment Model (FRAM), which analyzes CWT recoveries relative to a historical “base period” using a deterministic framework (Pacific Fishery Management Council, 2008; https://framverse.github.io/fram_doc/). The type of empirical, integrated model structure we present here represents a possible alternative or complementary method for fisheries modeling and forecasting, particularly if we can resolve ongoing challenges in model fitting. Our model quantifies uncertainty, which is neglected in most U.S. West Coast salmon management (Satterthwaite & Shelton, 2023), and is not dependent on the assumption that hatchery-origin fish are good proxies for nearby naturally-produced fish (Pacific Salmon Commission, 2007). This model framework may be particularly useful for generating metrics of pre-fishing abundance and distribution for future forecast models to capture.

Numerous supplemental data sources and possible model constraints have the potential to move the model closer to satisfactory convergence. First, obtaining independent estimates of broodyear-specific outgoing smolt or juvenile abundance would greatly facilitate model convergence, given the observed challenge in resolving estimates of initial stock abundance. Empirical estimates rarely exist for most stocks (there are exceptions however, such as daily smolt trap data collected by WDFW; Nelson et al., 2019; Anderson et al., 2020), but we envision stock-specific population dynamics models or stock-recruitment relationships could be utilized to generate estimates of smolt abundance. Second, systematic collection of age structure information for GSI sampled fish could be used to inform relative abundances of broodyears for specific stocks and improve model convergence. Specifying synthetic age structure constraints (i.e., constant among years, shared among stocks) based on a single stock-specific cohort reconstruction markedly improved model convergence in this application (i.e., CWT+GSI+Age), but we expect age structure to vary annually among stocks and years as a function of broodyear strength and environmental conditions. The age data ideally would be obtained using age readings based on either scales or otoliths. Data collection of this type, paired with GSI sampling, is rare, and the accuracy of scale age readings in particular is typically assessed and validated using only hatchery-origin fish (McNicol & MacLellan, 2010; Kormos, Palmer-Zwahlen & Miller, 2011; Harris, 2020). Alternatively, annual estimates of in-river age structure for spawning adults, either just for fish with CWTs or (ideally) all spawners, would allow us to obtain better estimates of age-specific return rates for each stock and further improve estimates of broodyear-specific ocean abundances. However, as is often the case, these data are rarely both systematically collected and publicly available (e.g., exceptions include data from Columbia River PIT tagging, Chulik et al., 2017; weir counts, Echave, 2009; PBT methodology, Beacham et al., 2021a). Similarly, comprehensive annual estimates of the expected total number of in-river CWTs for each CWT release group would be immensely helpful. These annual data, combined with the known starting release abundance, would improve estimates of fishery mortality rates over time and help resolve estimates of juvenile and fishery mortality that varied between models with and without GSI data. Detailed escapement data capable of generating these estimates exist for some stocks and years, although public availability varies (e.g., Kormos, Palmer-Zwahlen & Low, 2012). Some in-river data are available for many of the stocks included in the Pacific Salmon Commission’s Chinook salmon models (e.g., Pacific Salmon Commission, 2023), but none of the four stocks included here are included in those models and only some of the freshwater CWT recoveries used by the PSC are in the RMIS database (O. Shelton, 2023, pers. comm.). Finally, improved estimates of total run size for all stocks also would improve realized model convergence. We expect our input estimates of run size for rarer stocks (i.e., CAC, NCASOR) represent substantial under-estimates of true escapement based on inconsistent sampling methodology and survey implementation; this expectation is supported by the observation that model estimates of run size consistently exceeded input estimates for these stocks (Fig. S3.14).

Our approaches make a series of important assumptions about the biology of Chinook salmon and the connection between our observations and that biology. While we view our assumptions and model implementation as reasonable, we think it is important to be explicit about our approach for transparency and to allow future researchers to build on or depart from our approach with an understanding of our reasoning. First, in order to address data limitations and facilitate model convergence, we fixed a number of parameters to pre-specified values, including uncertainty for estimates of PFMC landings, stock-specific escapement age-structure, and run size estimates as well as constraints on spatial structure. For many of these parameters, we lacked the necessary data to directly estimate uncertainty within the existing model structure. Selecting alternative values for uncertainty parameters has the potential to modify model results by changing the confidence we have in competing data sources, while different spatial constraints may change fine-scale patterns in ocean distribution. Similarly, we again note that assuming spatial distributions of stocks did not vary annually fails to capture previously estimated differences in distribution as a function of temperature (Shelton et al., 2021); this assumption may have constrained expected variability in seasonal distributions and influenced estimation of other life history parameters. Our decision to reduce the spatial extent of the study to 8 ocean regions, spanning California to Washington, also could affect direct comparison of spatial distributions to those presented in previous studies (Shelton et al., 2019; Shelton et al., 2021). However, we found very few CWTs from modeled stocks are recovered north of Washington state (Table S2.5), and we expect model estimation and convergence would prove more difficult if we attempted to include all 17 regions with only sparse CWT data. Finally, the decision to model total release groups and subsequently relate catches and abundances associated with these groups to GSI observations added a great deal of complexity and challenge in achieving model convergence; however, given the existing structure of the CWT-only model, this approach represented the most feasible and logically consistent method of integrating the two data sources in a single framework. There were no clear connections between estimates of CWT release group abundances and GSI observations, in the absence of additional information on the ratio of tagged to untagged fish, as CWT release group abundances provide little information on overall stock abundances.

Conclusions

We developed and described the first completed model capable of integrating GSI and CWT data to estimate Chinook salmon stock distribution and abundance, building from a state-space model based on only CWT recoveries. Models resulted in new estimates of stock distribution and ocean abundance, using information from both hatchery- and natural-origin fish, for four fall-run Chinook salmon stocks from California and southern Oregon. Estimates of stock distribution represent the most comprehensive to date, especially for rarer stocks like California Coastal Chinook with limited records of CWT recoveries; California Coastal Chinook in particular are a threatened stock whose management poses ongoing challenges in the absence of sufficient data (O’Farrell et al., 2023). As expected, given the challenges of fitting complex integrated models, we did observe several challenges in model convergence that were not entirely resolved with supplemental data and constraints. Proposed modifications to fisheries data collection efforts have the potential to resolve these challenges. We expect these modeling advances will provide a useful foundation for future efforts to integrate CWT and GSI data and inform Pacific salmon management.

The described integrated model can be readily extended to inform the biology and management of other Chinook salmon stocks. Obtaining and incorporating GSI data from waters north of Washington (e.g., British Columbia, Alaska) would allow distribution and abundance modeling for all fall-run stocks in the Northeast Pacific Ocean (e.g., Guthrie III et al., 2017; Beacham et al., 2021b). Leveraging older GSI data sources based on protein electrophoresis and allozyme markers, instead of microsatellites or SNPs, may also increase temporal coverage of GSI data (e.g., Milner et al., 1985; Utter et al., 1987; Winans et al., 2001). Furthermore, the model can be expanded to include other Chinook salmon run types (e.g., spring-run, winter-run) by modifying modeled life history processes and assumptions. We anticipate the integrated modeling approach will be particularly valuable for these other run types, which generally occur at lower abundances and yield fewer CWT recoveries. Introducing modeling of environmental covariates like temperature also can allow for inter-annual variability in stock distributions and help clarify whether differences in estimated distributions between models with and without GSI data can be attributed to temporal differences in sampling efforts. These advances and modifications will improve the value of this model to ongoing fisheries management, which already represents a comprehensive framework for integrating numerous data sources and providing robust estimates of stock distribution and abundance.

Supplemental Information

Supplemental Information 1 Full model description

Click here for additional data file.

Supplemental Information 2 Summary of updated and new data sources

Click here for additional data file.

Supplemental Information 3 Model diagnostics

Click here for additional data file.

Supplemental Information 4 Summary of coded wire tag (CWT) unique tag codes used to identify focal CWT recoveries for modeling

Click here for additional data file.

Supplemental Information 5 Codebook for data

Click here for additional data file.

We would like to thank all reviewers for their comments and improvements to this work.

Additional Information and Declarations

Competing Interests

Author Contributions

Data Availability

Eric. J. Ward is an Academic Editor for PeerJ. The authors declare that they have no other competing interests.

Alexander J. Jensen conceived and designed the experiments, performed the experiments, analyzed the data, prepared figures and/or tables, authored or reviewed drafts of the article, and approved the final draft.

Ryan P. Kelly conceived and designed the experiments, authored or reviewed drafts of the article, and approved the final draft.

William H. Satterthwaite conceived and designed the experiments, analyzed the data, authored or reviewed drafts of the article, and approved the final draft.

Eric J. Ward conceived and designed the experiments, analyzed the data, authored or reviewed drafts of the article, and approved the final draft.

Paul Moran conceived and designed the experiments, analyzed the data, authored or reviewed drafts of the article, and approved the final draft.

Andrew Olaf Shelton conceived and designed the experiments, performed the experiments, analyzed the data, prepared figures and/or tables, authored or reviewed drafts of the article, and approved the final draft.

The following information was supplied regarding data availability:

All described model data, code, and outputs are available at Zenodo: Jensen, A., Kelly, R., Satterthwaite, W., Ward, E., Moran, P., & Shelton, A. O. (2023). Chinook Salmon GSI CWT Integrated Model (1.0.0). Zenodo. https://doi.org/10.5281/zenodo.8057795.

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
