# Peer review of "Modeling ocean distributions and abundances of natural- and hatchery-origin Chinook salmon stocks with integrated genetic and tagging data"

_PeerJ, doi:10.7717/peerj.16487_

## Round 0.1 · original submission · Major Revisions

Your article is very interesting, but I suggest that you correct it in order to improve the translation from the language of models to the language of conservation for a non-specialist audience, which I believe will increase the interest and impact of the manuscript.

Reviewer 1 ·

Basic reporting

Methods need some clarification

Experimental design

Nice approach to important problem

Validity of the findings

Good discussion of results and limitations

Additional comments

Summary: This manuscript develops a methodology to combine coded-wire tag recovery data with genetic stock proportion estimates to more powerfully estimate ocean abundance and spatial distribution of salmon stocks. The test case is a set of four stocks from Oregon and California. In applying the methodology, convergence issues were experienced which were addressed by including additional priors/constraints.
General thoughts: This attempt to meld CWT and GSI data to improve salmon abundance and distribution estimates is a stab at making valuable improvements to the methods used for assessing the stocks supporting valuable fisheries up and down the US and Canadian Pacific Coast. In the Discussion, the authors do a good job of interpreting their results, describing the difficulties they encountered and suggesting future improvements. Overall, the approach is solid, but might benefit from rethinking some aspects, and the manuscript could use a re-edit to improve clarity of methods. I’m confident the authors can bring the manuscript to a publishable state, although I wonder if they might consider waiting until they settle on approaches that give good performance. Just a suggestion – publishable as a progress report.
Major comments:
1. The methods are not quite replicable. The descriptions of data and analyses in Methods and in S1-2 are sometimes vague. I’m unclear why the description of the data and the likelihood functions used are relegated to appendices, with only vague language in the Methods in the main body. It would be helpful to have common terminology and section headings in Methods and Supplements.
Specific areas of vagueness include:
a) S.2 starts with a description of effort data for all fisheries, but it’s not clear whether these data are used. F values for fisheries don’t seem to be described as a function of effort in S.1, so it’s unclear how/whether these data are used.
b) “Exploration of Synthetic Age Structure” in the Methods is very hard to follow, and unnecessarily vague. The description in App. S.1 is much more specific and clear. Synthetic age structure is just a set of multinomial priors on the age structure in the ocean, and it would be clearer to refer to it in those terms.
2. The Discussion outlines the difficulties with estimating initial release sizes for groups which results in convergence issues for the combined CWT/GSI model. Two other potential contributors to the convergence issues when combining the CWT and GSI data are:
- contradictory data = the two data sources imply different distributions and abundances
- little data overlap = in Fig. 2, the two stocks on the right have weak overlap between the GSI and CWT data, so intercalibration might be difficult
3. “Synthetic age structure” is just a set of priors on the age structure in the ocean. Unfortunately, even though they come from a different analysis than the one in the manuscript (PFMC 2021), they’re based on the same CWT data. This violates a basic principle of prior construction, that it be based on external information, leading to double-weighting of the information used in the modeling and an overestimate of estimation precision.
4. This is also a case where the different descriptive headings (“Synthetic age structure” in Methods, “Specifying age-structure data for total release abundances”) hinders the reader. The section of the Discussion on this model also shifts among terminologies. Although the basis of the assumed age averages for three stocks is (vaguely) sourced (PFMC 2021 from Klamath River values), there is no explanation for how the differing values for the younger SFB stock were chosen.
5. The method for dealing with salmon escapement proportions (S.1 Observation models and data for Chinook salmon escapement proportions) assumes escapement abundance isn’t accurate, but implicitly assumes the bias is constant. Otherwise, the release group brood year CWT counts from different years of escapement wouldn’t be comparable. This assumption was made previously by Shelton, but still deserves some discussion of why it might be valid.
6. I would have liked more discussion of the systematic biases in observed and predicted stock proportions in the CWT+GSI results in Figs. S3.11-13 and the temporal pattern in S3.14.
Minor issues:
S1.2 description: “M and F and U”?
S.2 starts with a description of effort data for all fisheries, but it’s not clear whether these data are used. F values for fisheries don’t seem to be described as a function of effort in S.1.
S3.5-6 – give plots descriptive headers, e.g., “Spring-2s”
S3.9 & 20. There are no top and bottom panels. And if taken literally, it says a 16 inch fish that is 40 months old is 5x as vulnerable to hook and line gear as a 16 inch fish that is 10 months old. Something is wrong with this figure.
S3.14 – bigger symbols for clarity

Reviewer 2 ·

Basic reporting

No comment

Experimental design

No comment

Validity of the findings

No comment

Additional comments

General – This is an excellent and rigorous effort to integrate disparate data sources for an enhanced understanding on the distribution, abundance, and life history parameters of Chinook salmon in the NE Pacific. The technical discussion needs few minor edits but I ask that the authors consider emphasizing the value of their advancements in the broader conservation context, which would help convince non-specialist readers of the value of this study and potential value of continuing such integrated modelling efforts into the future. With the suggestions for future sampling programs (such as matched GSI and age) that will cost taxpayer money, this translation from model-speak to conservation-speak will be important. Notably, I think further discussion of the implications of higher juvenile and fishing mortality inferred by the CWT+GSI model is warranted given their major implications on the effectiveness of conservation efforts (at preceding or later life stages) to maintain population viability (e.g., dam removal to improve smolt survival during downstream migration or in-river protections for returning adults). The relative proportion of overall mortality contributed during the juvenile at-sea or returning adult fishery stages, and if these are first-order mortality bottlenecks, should strongly constrain long-term horizons for recovery of imperiled stocks. Figures S3.7 and 3.18 suggest a near doubling of juvenile mortality rate, for example, for which it seems very important to emphasize the degree of difference. Discussions of salmon-directed conservation projects, such as the proposed Snake River dams removal, come to mind amidst this issue of where mortality bottlenecks are occurring and most affecting imperiled populations; this, in turn, should influence the effectiveness of conservation efforts at non-bottleneck points and, thus, whether their ecological returns are worth the potential financial, social, and community costs relative to other options.

Line 187 – What is the potential sensitivity of this analysis to this assumption, given the likely variation in spatial distributions from year-to-year?

Line 225-226: Is there a sense of sampling similarity for GSI between recreational and commercial landings? For example, in a given year, how do these two sources compare? I realize this may not be known, but if any comparison is available, I suggest highlighting this similarity (or potential lack thereof).

Line 255-266: This weighting occurs as a function of the degree of overdispersion, with more overdispersed data being weighted less?

Line 455-457: Explicitly clarify what parameters Age information inclusion does change, e.g., juvenile at-sea mortality.

Line 473-476: Might be worthwhile to reiterate your assumption of constant interannual spatial distribution.

Line 476-479: This addresses my earlier point on their comparability. Is there any trend in the bias between recreational and commercial that can quickly be summarized in a clause here (or preferably in the methods)?

Line 554-557: Do the ‘utilized’ estimates refer to the input estimates or model estimates? Wording here gets a bit confusing.

Line 585-590: Maybe give an example of a major known issue in use of maximum assignment probability to clarify for readers unfamiliar with such problems.

Supplement – Ensure all figure labels have appropriate units detailed. For example, Fig. S3.7 does not clarify whether juvenile mortality rate is daily or another unit of time.

·

Basic reporting

No comment

Experimental design

No comment

Validity of the findings

No comment

Additional comments

I truly appreciate the opportunity to review this paper. As someone who works fairly closely to Chinook salmon fisheries management, I see this paper as extremely relevant and responsive to needs for effective fisheries management. The paper is clearly written and fully explains the methodology and results. The introduction does a good job of introducing the paper and the discussion flows logically from the rest of the paper. I think this paper is suitable for publication and I provide some minor suggestions below.

General comment
I’m not sure but I wonder if the link between this model and current management methods (e.g., harvest control rules) could be strengthened and if that might add to the paper. Would this model represent an alternative way to estimate stock-specific catch and potentially address some shortcomings of existing methods, or is the primary use from a management perspective that it estimates the abundance of CS present in different areas at different times of year? I am still wrapping my head about the Pacific Salmon Treaty and PFMC management, but I feel like some of the harvest control rules (e.g., PST ASBM fisheries) are based on the abundance of CS in particular marine areas, which this model estimates. It may be that most readers already understand the value of estimating the different quantities that this model does, but I don’t think it would hurt to make it really explicit. It’s a very complex model, but I suspect it is all necessary to estimate quantities of interest. Perhaps the estimates of maturation rates would be very useful for calculating the adult equivalent (AEQ in management jargon) of catch of immature fish in the ocean. Ok, I see now that there is some of this already on line 512-519. This is helpful and I would not be opposed to having even more if the authors are so inclined.

Line comments

Line 140-141: This is all we could ask of any paper and this one represents a substantial and timely contribution.
Line 434: I think you say why these are “the most comprehensive” estimates to date above, but it might not hurt to reiterate what makes this so.
Line 476: I recommend adding a clause elaborating how it would be technically difficult, or just omitting the part of the sentence saying it would be difficult.
Line 481: “distributions” is repeated.
Line 526: I agree that having an underlying stock recruitment relationship could help constrain things.
Line 579-582: A potential consideration for future work is that perhaps one could make use of the proportion of CWT-tagged fish out of all of the adipose clipped fish from a hatchery release (so they can be differentiated from wild fish about which data on juvenile abundance might not be available). Obviously, we don’t know the hatchery release of the ad-clipped-no-CWT fish without GSI, so I’d have to think about it more, but maybe those data on releases of untagged hatchery juveniles from RMIS would be useful in some way or another.

Supplement S1

Equation S1.7: Consider providing the definition of N_i,a,l,s here as “the number of fish remaining in the ocean in each region at the end of the fall (S1.4).” given that it was defined in a previous section and it isn’t immediately obvious that it refers to a very specific abundance (or at least it took me a minute to figure out).
Bottom of page 8 top of page 9: Where it says “The estimated number is a deterministic expansion of the actual observed number of fish (most commonly 1) by the fraction of the fishery catch sampled for CWT (generally between about 10% and 50%; Supplement S2).”, consider defining the terms used to represent those quantities (O_(j,a,l,g), C_(i,a,l,g), and ϱ_(j,l,c,s,g)). I was a little confused when it was described first with no math and then again with math because I wasn’t sure if it was actually the same calculation being described (which I think it is).
Second to last sentence in second paragraph on page 9: I’m just curious if it would be possible to calculate an average ϱ_(j,l,c,s,g) across j for use in the model as a weighted mean or something based on the number of recovered tags at each sampling rate, as an alternative to using the median.
Expected and observed CWT recoveries observation components: This is just really cool. That is all.
General comment: Thinking about using this model for management, I wonder how managers would feel about estimating harvest as a latent variable as opposed to fixing it. I am aware that the values that it would be fixed at are just estimates themselves, but maybe the optics are better if harvest were to at least be reported with uncertainty based on whatever sampling and calculations are done, and then that uncertainty determines the CV of the observation error in the model.
End of page 14: I’m sure you thought of this, but it seems like the bits about the hierarchical structure of the proportion of focal stocks in landings could go in the previous section, “Observation models and data for Chinook salmon run sizes”.
Table S1.2: Remove parameters about effects of SST on ocean distribution.
Table S1.2: Confirm that κ_1and κ_2 parameters are in fact used in this model version.
Table S1.2: Finish definition of f_i

---

## Round 0.2 · accepted · Accept

I am pleased to confirm that your paper has been accepted for publication in PeerJ.

Thank you for submitting your work to this journal.

With kind regards,

Reviewer 1 ·

Basic reporting

Addressed confusing text highlighted by reviewers

Experimental design

Addressed ambiguities identified by reviewers

Validity of the findings

Good interpretation of strengths and limitations

Additional comments

The authors carefully considered my comments and responded appropriately

·

Basic reporting

no comment

Experimental design

no comment

Validity of the findings

no comment